# Eastward-propagating planetary wave in the polar middle atmosphere

Liang Tang[1], Sheng-Yang Gu[1*], Xian-Kang Dou[1]

[1] Electronic Information School, Wuhan University, Wuhan, China.

*Corresponding author: Sheng-Yang Gu, (gushengyang@whu.edu.cn)

**Abstract.** According to MERRA-2 temperature and wind datasets in 2019, this study presented the global variations of the eastward propagating wavenumber 1 (E1), 2 (E2), 3 (E3) and 4 (E4) planetary waves (PWs) and their diagnostic results in the polar middle atmosphere. We clearly demonstrated the eastward wave modes exist during winter periods with westward background wind in both hemispheres. The maximum wave amplitudes in the southern hemisphere (SH) are slightly larger and lie lower than those in the northern hemisphere (NH). Moreover, the wave perturbations peak at lower latitudes with smaller amplitude as the wavenumber increases. The period of the E1 mode varies 3-5 days in both hemispheres, while the period of E2 mode is slightly longer in the NH (~48 h) than in the SH (~40 h). The periods of the E3 are ~30 h in both SH and NH, and the period of E4 is ~24 h. Despite the shortening of wave periods with the increase of wavenumber, their mean phase speeds are relatively stable, which are ~53 m/s, ~58 m/s, ~55 m/s and ~52 m/s at 70˚ latitudes for W1, W2, W3 and W4, respectively. The eastward PWs occur earlier with increasing zonal wavenumber, which agrees well with the seasonal variations of the critical layers generated by the background wind. Our diagnostic analysis also indicated that the mean flow instability in the upper stratosphere and upper mesosphere might contribute to the amplification of the eastward PWs.

## 1 Introduction

The dominance of large amplitude planetary waves in the stratosphere, mesosphere and lower thermosphere regions and their interactions with zonal mean winds are the primary driving force of atmospheric dynamics. In addition, sudden stratospheric warmings (SSWs) and quasi-biennial oscillation (QBO) events can dynamically couple the entire atmosphere from the lower atmosphere to the ionosphere (Li et al., 2020; Yamazaki et al., 2020; Yadav et al., 2019; Matthias and Ern, 2018; Stray et al., 2015). Westward propagating planetary wave is one of the prominent features during austral and boreal summer. Westward quasi-2-day waves (Q2DWs) are the most obvious representative waves and one of the most investigated phenomena using planetary wave observations. Most of the previous studies focused on the westward propagating Q2DWs, i.e., zonal wavenumbers of 2 (W2), 3 (W3) and 4 (W4) modes (Lainer et al., 2018; Gu et al., 2018b; Wang et al., 2017; Pancheva et al., 2016; Gu et al., 2016a; Gu et al., 2016b; Lilienthal and Jacobi, 2015; Gu et al., 2013; Limpasuvan and Wu, 2009; Salby, 1981). However, limited studies were conducted to understand the seasonal variations of the occurrence date, peak amplitude and wave period for the eastward Q2DWs (Gu et al., 2017; Lu et al., 2013; Alexander and Shepherd, 2010; Sandford et al., 2008; Palo et al., 2007; Merzlyakov and Pancheva, 2007; Manney and Randel, 1993; Venne and Stanford, 1979).

Typically, Q2DWs maximize after the summer solstice in the middle latitudes. The largest wave amplitudes generally appear near the mesopause in January–February in the Southern Hemisphere (SH), while in the Northern Hemisphere (NH) in July–August (Tunbridge et al., 2011). W3 and W4 Q2DWs reach amplitudes during austral and boreal summer in the mesosphere and lower thermosphere, respectively. The seasonal variation of westward Q2DWs activity is obvious (Liu et al., 2019; Gu et al., 2018b; Rao et al., 2017). By observing the long-term Q2DW in the NH and SH, Tunbridge et al. (2011) reported that W3 is generally stronger than the other two modes in the SH, reaching the amplitude of ~12K; while W4 is stronger than W3 in the NH, reaching ~4K. Moreover, W4 generally lives longer than W3, and W4 can still be observed after the ending of W3. A previous study has demonstrated that wave source, instability, critical layer and mean zonal wind are the primary reasons for the seasonal variation of Q2DWs (Liu et al., 2004). By studying the long-term satellite datasets in the SH, Gu et al. (2019) have suggested that the strongest events of W2, W3 and W4 could be delayed with increasing the zonal wavenumber, and these events would be indistinguishable during SSWs. The wave periods of W4, W3 and W2 vary around ~41-56 h, ~45-52 h, and ~45-48 h, respectively. Furthermore, W2 can be observed using global satellite datasets, but it has an amplitude weaker than W3 and W4 in the NH and SH (Meek et al., 1996). The propagation and amplification of

Q2DWs are primarily modulated by instability, refractive index and critical layer, while the variation of background wind may cause different zonal wavenumber events (Gu et al., 2016a; Gu et al., 2016b). By analyzing the variation of Q2DWs activity during SSWs, Xiong et al. (2018) noticed that W1 is generated by the nonlinear interaction between SPW2 and W3. During SSWs, the coupling between the NH and SH can enhance the summer easterly and promote the nonlinear interaction between W3 and SPW1 (Gu et al., 2018b).

Some recent studies have discovered significant eastward planetary waves in the polar stratosphere and mesosphere regions, with periods of nearly two and four days (Gu et al., 2017; Sandford et al., 2008; Merzlyakov and Pancheva, 2007; Coy et al., 2003; Manney and Randel, 1993). Planetary waves with zonal wavenumbers -1 (E1) and -2 (E2) correspond to 4- and 2-day waves, respectively. Furthermore, planetary waves of 1.2-day with wavenumber -3 (E3) and 0.8-day with wavenumber -4 (E4) have been reported to contain the same phase speeds as E1 and E2 (Manney and Randel, 1993). This series of eastward planetary wave can significantly affect the thermal and dynamic structure of the polar stratosphere, resulting in profound changes in the wind and temperature of the polar stratosphere (Coy et al., 2003; Venne and Stanford, 1979). Beyond the knowledge about nonlinear interactions between migrating tides and Q2DWs (Palo et al., 1999), further investigation has confirmed

that E2 Q2DW could be generated by the nonlinear interaction between planetary wave and tides in the mesosphere and lower thermosphere (MLT) (Palo et al., 2007). We should note that the E2 Q2DW generated in the MLT region is different from that in the polar stratosphere, which is discussed in this paper.

By studying and analyzing satellite datasets, Merzlyakov and Pancheva. (2007) indicated that the wave periods of E1 and E2 events range 1.5-5 days. They reported that EP flux travels from the upper to the lower atmosphere, meaning that the upper atmosphere has a dynamic influence on the lower atmosphere. Sandford et al. (2008) reported about significant fluctuations of E2 Q2DW in the polar mesosphere. They indicated the influence of changes in mean zonal winds during a major SSW on the propagation of polar E2. In addition, they proposed the significance of E2 fluctuation in the mesosphere driven by the instabilities in the polar night jet. For E2, amplitude of temperature, zonal wind and meridional wind during the austral winter can reach ~10 K, ~20 m/s and ~30 m/s, respectively; while those during the boreal winter can drop by almost two-thirds. Lu et al. (2013) found that eastward planetary wave propagation is limited to the winter high latitudes probably because the negative refractive indices equatorward of ~45°S result in evanescent wave characteristics. That study suggested that the instability region at ~50-60°S might be induced by the stratospheric polar night jet and/or the "double-

jet" structure.
In this study, we use the second modern retrospective research and
application analysis (MERRA-2) datasets to investigate the eastward
propagating wave characteristics of the stratosphere and mesosphere in
polar region in 2019, including E1, E2, E3 and E4. Specifically, we
investigate the variation of the occurrence date, peak amplitude and wave
period of eastward waves; as well as the role of instability, background
wind structure and critical layer in the propagation and amplification of
eastward waves. The remaining parts of this paper are organized as follows.
Section 2 describes the data and methods used in this study. Section 3
analyzes the global latitude-temporal variation structure of eastward waves
during winter in 2019. The amplification and propagation features of the
eastward planetary waves in the NH and SH with different wavenumber
events are examined in Sections 3.1 and 3.2, respectively. Section 3.3
compares and analyzes the eastward waves in the NH and SH. All research
results are summarized in Section 4.
**2 Data and Analysis**
To extract the E1-, E2-, E3- and E4-wave, we apply the least-square
method to each time window (e.g., 10-day, 6-day, 4-day and 4-day), and
then use time window to determine the amplitude. (Gu et al., 2013). This
method has been shown to successfully identify planetary waves from
satellite measurements (Gu et al., 2019; Gu et al., 2018a; Gu et al., 2018b;
Gu et al., 2018c; Gu et al., 2013).
$$y = A\cos[2\pi(\sigma \cdot t + s \cdot \lambda)] + B\sin[2\pi(\sigma \cdot t + s \cdot \lambda)] + C \qquad (1)$$
The least-squares method is used to fit the a set of parameters ($A$, $B$
and $C$), where $\sigma$, $t$, $s$ and $\lambda$ are the frequency, UT time, zonal
wavenumber and longitudes. The amplitude of wave $R$ can be expressed
as $R = \sqrt{A^2 + B^2}$.
The second modern retrospective research and application analysis
(MERRA-2) covers the long-term atmospheric reanalysis datasets initiated
by NASA in 1980. It has been upgraded recently using the Goddard Earth
Observing System Model, Version 5 (GEOS-5) data assimilation system.
Briefly, MERRA-2 includes some updates to the model (Molod et al., 2014;
Molod et al., 2012) and the Global Statistical Interpolation (GSI) analysis
scheme of Wu et al. (2002). The MERRA-2 data consist of various
meteorological variables, e.g., net radiation, temperature, relative humidity
and wind speed. The spatial coverage of MERRA-2 data is the globe
(spatial resolution: 0.5°×0.625°; temporal resolution: 1 h). This
meteorological data are widely used to detect the middle atmosphere such
as the planetary wave in the polar atmosphere, global thermal tides, climate
variability and aerosol (Ukhov et al., 2020; Sun et al., 2020; Bali et al.,
2019; Lu et al., 2013). Many recent studies indicated the feasibility of using
MERRA-2 data for the kind of research in present study. Therefore, we
apply the MERRA-2 datasets to obtain the variation in background wind,
instability, refractive index and critical layer; and explore the patterns of
eastward planetary waves propagation and amplification through
diagnostic analysis.
The critical layer will absorb or reflect planetary waves from the lower
atmosphere during upward propagation. Planetary waves that gain
sufficient energy in the unstable region will be amplified during reflection.
In a sense, the critical layer plays an important role in regulating the
amplification and propagation of planetary waves (Gu et al., 2016a; Gu et
al., 2016b; Liu et al., 2004).
$$\overline{q}_{\varphi} = 2\Omega\cos\varphi - \left(\frac{\left(\overline{u}\cos\varphi\right)_{\varphi}}{a\cos\varphi}\right)_{\varphi} - \frac{a}{\rho}\left(\frac{f^2}{N^2}\rho\overline{u}_z\right)_z \qquad (2)$$
The baroclinic/barotropic instability in the atmospheric space
structure is caused by the simultaneous equalization of the negative latitude
gradient and the quasi-geostrophic potential vorticity ($\overline{q}_{\varphi}$). In Equation (2),
$\Omega$ is the angular speed of the Earth's rotation; $\varphi$ is the latitude; $\overline{u}$ is the
zonal mean zonal wind; $a$ is the Earth radius; $\rho$ is the air density; $f$ is the
Coriolis parameter; $N$ is the buoyancy frequency; subscripts $z$ and $\varphi$ are the
vertical and latitudinal gradients.
According to Andrews et al. (1987), the properties of planetary wave
propagation can be calculated using the Eliassen-Palm (EP) flux vectors
(F), i.e.,

$$F = \rho a \cos\varphi \begin{bmatrix} \overline{u_z}\dfrac{\overline{v'\theta'}}{\overline{\theta_z}} - \overline{v'u'} \\[2ex] \left[ f - \dfrac{\left(\overline{u}\cos\varphi\right)_\varphi}{a\cos\varphi} \right] \dfrac{\overline{v'\theta'}}{\overline{\theta_z}} - \overline{w'u'} \end{bmatrix} \tag{3}$$

where $u'$ and $v'$ are the planetary wave perturbations in the zonal and meridional wind, respectively; $\theta'$ and $w'$ are the potential temperature and vertical wind, respectively. The planetary wave propagation is only favorable where the square of refractive index $m^2$ is positive:

$$m^2 = \frac{\overline{q_\varphi}}{a\left(\overline{u}-c\right)} - \frac{s^2}{\left(a\cos\varphi\right)^2} - \frac{f^2}{4N^2H^2} \tag{4}$$

where $s$ is the zonal wavenumber, $c$ is the phase speed and $H$ is the scale height. The square of the refractive index is taken as the waveguide of planetary waves, i.e.,

$$c = -\upsilon_0 \cos\left(\frac{\varphi\pi}{180}\right)\bigg/ sT \tag{5}$$

where $\upsilon_0$ is the equatorial linear velocity, $s$ is the zonal wavenumber and $T$ is the wave period.

**3 Results and Discussion**

Figure 1 shows the global temporal-latitude variation structures of E1, E2, E3 and E4 extracted from the 2019 MERRA-2 temperature datasets using time windows 10-, 6-, 4- and 4-days, respectively. The mean temperature amplitude of E1, E2, E3 and E4 at 55.4 km during the periods 3~5-, 1.5~2.5-, 1~1.5- and 0.9~1.1-day are displayed in Figure 1a, 1b, 1c and 1d, respectively. Eastward waves are characterized by obvious

seasonal variations in the SH and NH. In addition, E1, E2 (E3) and E4 reach their maximum amplitude at 50-80˚(S/N). In the SH, the strongest E1 and E2 events occur on days 209-218 and 167-172; while E3 and E4 events occur on days 151-154 and 139-142. This means that their occurrence date of maximum amplitude gets earlier with increasing zonal wavenumber. In addition, the maximum amplitude of E1, E2, E3 and E4 are ~6.0 K, ~4.2 K, ~3.6 K and ~2.4 K, respectively, indicating that their peak amplitude drop with rising zonal wavenumber. In the NH, the strongest E1, E2, E3 and E4 events occur on days 41-50, 69-74, 35-38 and 63-66, respectively; the corresponding peak amplitude are ~5.5 K, ~3.8 K, ~2.8 K and ~1.2 K, respectively. Whilst the results demonstrate the decline of the peak amplitude with increasing zonal wavenumber in the NH, the occurrence date is irregular. Moreover, E4 is relatively weak in the NH and difficult to find, so W4 is insignificant in the NH. Figure 2 presents the changes in zonal mean zonal wind at 70˚S and 70˚N in 2019. It can be seen that the background wind on days 90-240 (70˚S) is dominated by westward wind, and reaches ~80 m/s at ~50 km on days 210; it is dominated by eastward wind in late and early 2019, and reaches ~-40 m/s at ~60 km. Meanwhile, the background wind is primarily westerly wind in late and early 2019 (70˚N), and reaches ~90 m/s at ~60 km on days 50; while on days 120-240, the background wind is primarily easterly wind, and the amplitude reaches -40 m/s on days 200. Compared with Figure 1, the

results show that the eastward wave modes exist during winter periods with westward background wind in both hemispheres.

**3.1 In the Southern Hemisphere**

Figure 3 shows that observed maximum temperature amplitude is at ~48.2 km and ~70-80°S for E1; ~48.2km and ~60-70°S for E2 and E3; ~48.2km and ~50-60°S for E4. For E1, the observed maximum perturbation occurs on days 211-220 (with an amplitude of ~8.5 K), and the remaining fluctuations occur on days 161-170, 187-196 and 231-240. For E2, the observed maximum perturbation happens at days 219-224 (with an amplitude of ~7.8 K), and three peaks appear on days 139-144, 173-178 and 187-192. Regarding E3, the strongest perturbation occurs on days 151-154 (with an amplitude of ~5.2 K), while the rest are distributed on days 141-144, 201-204 and 209-202. E4 perturbations are distributed on days 127-130, 145-148, 161-164, 213-216, with weak amplitude of ~3 K. Since earlier studies mentioned that the wave period of the eastward wave can vary, we also investigate the periodic variabilities of E1, E2, E3 and E4. The results show that the period corresponding to the maximum perturbation of E1 falls between ~106 (days 187-196) and ~69 h (days 211-220), and their wave periods vary significantly. Nonetheless, the wave period of E2 gradually changes from ~42 h (days 139-144) to ~38 h (days 219-224), and its stability is stronger than that of E1. The wave periods of E3 and E4 are about ~39 h and ~24 h, respectively. These results reflect

that E2, E3 and E4 wave periods are more stable compared to E1.

The spectra, spatial (vertical and latitudinal) structures of temperature,

zonal and meridional wind, and diagnostic analysis of E1 are extracted
from the two corresponding events (refer to Figure 4). Figures 4a, 4b show
the least-squares fitting spectra for MERRA-2 temperature on days 187–
196, 211-220 at ~48.2 km and ~70-80°S, when and where the E1
maximizes. An eastward wavenumber -1 signal with the periods of ~106 h
and ~69 h clearly dominates the whole spectrum. The temperature spatial
structures corresponding to these E1 (i.e., ~106 h and ~69 h) are displayed
in Figures 4c, 4d. The temperature spatial structure of E1 exhibits obvious
amplitude bimodal structure at ~70-80˚S and ~50 km, and ~70-80˚S and
~60 km, with the maximum at ~70-80˚S and ~50 km. The strongest
temperature amplitude of E1 occurs at ~50 km and ~70-80˚S with an
amplitude of ~10 K on the days 211-220, and the other peak is ~9K (~70-
80˚S and ~60 km). The temperature amplitude of ~9K occurs at ~50 km
and ~70-80˚S during days 187-196, and the rest is ~7K (~70-80˚S and ~60
km). The corresponding spatial structures of zonal wind and meridional
wind of these E1 are shown in Figures 4e, 4f, 4g and 4h. The maximum
zonal wind amplitude of E1 occurs at ~60-70˚S and ~60 km with an
amplitude of ~14 m/s on days 187-196, and ~20 m/s at ~50-60˚S and ~60
km on days 211-220. The amplitude of E1 meridional wind hits ~10 m/s at
~70-80˚S and ~55 km (days 187-196) and ~17 m/s at ~70-80˚S and ~60
km (days 211-220), respectively.

Figures 4i, 4j show the diagnostic analysis results for the E1 events

during days 187–196 and 211–220, respectively. Apparently, the EP flux
vectors is more favorable to propagate in the SH winter and is dramatically
amplified by the mean flow instabilities and appropriate background winds
at polar region and between ~40 km and ~80 km, with EP flux propagating
into the upper atmosphere (Figure 4i). Meanwhile, there is an EP flux at
the mid-latitudes and ~60-80 km, which propagates into the lower
atmosphere. The wave-mean flow interaction near its critical layer (106 h)
of the green curve amplifies E1, and the positive refractive index region
surrounded by the yellow curve also enhances E1 propagation. In addition,
the strong instability and weak background wind at ~70-80°S and ~40-60
km provide sufficient energy for the upward EP flux to propagate and
amplify. Nevertheless, the downward propagating EP flux is amplified by
weak instability and strong background wind at ~50-60°S and ~60-70 km.
Besides, both upward and downward EP fluxes eventually propagate
toward the equator at ~50 km. Figure 4j shows that EP flux on days 211-
220 propagates downward and amplifies after the interaction of the critical
layer (~69 h). The positive refractive index region, strong instability and
weak background wind at ~50-60°S and ~60-70 km provide sufficient
energy for E1 amplification and propagation, and ultimately point towards
the equator at ~50 km. The results show that the weak background wind

and strong instability in the polar region can promote the upward propagation and amplification of EP flux. Meanwhile, the appropriate background wind and instability in the mid-latitudes are also conducive to the downward propagation and amplification of EP flux. In other words, instability and appropriate background wind dominate the propagation and amplification of E1.

For E2, the spectra are observed at ~48.2 km and ~60-70°S on days 173-178 and 219-224 when the eastward wavenumber -2 becomes the primary wave mode with the wave periods ~38 h and ~39 h, respectively (as shown in Figures 5a, 5b). The temperature spatial structures corresponding to these E2 (~38 h and ~39 h) are presented in Figures 5c, 5d. The temperature spatial structure of E2 shows an obvious amplitude bimodal structure at ~60-70˚S and ~50 km, and ~60-70˚S and ~60 km, with the maximum at ~60-70˚S and ~50 km. The maximum temperature amplitude of E1 occurs at ~50 km and ~60-70˚S with an amplitude of ~7.5 K on the days 173-178, and the other peak is ~6 K (~70˚S and ~60 km). The temperature amplitude of ~10 K happens at ~50 km and ~60-70˚S during days 219-224, and the rest is ~6 K (~70˚S and ~60 km). The corresponding spatial structures of zonal wind and meridional wind of these E2 are illustrated in Figures 5e, 5f, 5g and 5h. The zonal wind spatial structure of E2 shows an obvious amplitude bimodal structure at ~50-60˚S and ~60 km, and ~70-80˚S and ~60 km, with the maximum at ~50-60˚S

and ~60 km. The maximum zonal wind amplitude of E2 appear at ~50-60˚S and ~60 km with an amplitude of ~10 m/s on days 173-178, and the other peak is ~9 m/s (~70-80˚S and ~60 km). The zonal wind amplitude of ~20 m/s occurs at ~50-60˚S and ~60 km on days 219-224, and the rest is ~15 m/s (~70-80˚S and ~60 km). The amplitude of E2 meridional wind reaches ~13 m/s at ~70-80˚S and ~60 km (days 173-178) and ~27 m/s at ~70-80˚S and ~60 km (days 219-224), respectively.

Figures 5i and 5j illustrate the diagnostic analysis during days 173-178 and 219-224 for E2, respectively. Obviously, E2 is more likely to propagate in the SH winter and is dramatically amplified by the mean flow instabilities at the middle-high latitudes between ~40 km and ~80 km. With EP flux propagating into the lower atmosphere, it eventually propagates toward the equator at ~50 km. Besides, E2 is amplified and propagated by the wave-mean flow interactions near its critical layer (~38 h) of the green curve, and the promoting effect of the positive refractive index region surrounded by the yellow curve. Meanwhile, the weak instability and strong background wind at ~50-60˚S and ~50-70 km provide the energy for the propagation and amplification of EP flux into the lower atmosphere during days 173-178 (Figure 5i). According to the diagnostic analysis of days 219-224, E2 obtains sufficient energy from strong instability and strong background wind at ~50-60˚S and ~60-70 km. It is amplified and propagated into the lower atmosphere through the critical layer and

positive refractive index action (as shown in Figure 5j). The results show
that the background wind at ~50-60˚S and ~50-70 km is weaker on days
173-178 than on days 219-224; and the instability at ~50-60˚S and ~60-70
km is stronger on days 219-224 than on days 173-178. Our results show
that E2 has absorbed sufficient energy to be amplified under the
background conditions during days 219-224 (Figures 5a, 5b).

Figures 6a and 6b show the observed spectra of E3 at ~48.2 km and

~60-70°S on days 151-154 and 201-204, and the wave periods of locked
wavenumber -3 are ~29 h and ~29 h, respectively. The corresponding
temperature spatial structures of these E3 (i.e., ~29 h and ~29 h) are
displayed in Figures 6c, 6d. The temperature spatial structure of E3 shows
an obvious amplitude bimodal structure at ~60-70˚S and ~50 km, and ~60-
70˚S and ~60 km, with the maximum at ~60-70˚S and ~50 km. Besides,
E3 also has a weak peak at ~60-70˚S and ~70 km. The strongest
temperature amplitude of E3 occurs at ~50 km and ~60-70˚S with an
amplitude of ~6K on the days 151-154, and the other peak is ~5 K (~60-
70˚S and ~60 km). The temperature amplitude of ~5 K happens at ~50 km
(~60 km) and ~60-70˚S during days 201-204. The corresponding spatial
structures of zonal wind and meridional wind of these E3 are shown in
Figures 6e, 6f, 6g and 6h. The zonal wind spatial structure of E3 shows an
obvious amplitude bimodal structure at ~70-80˚S and ~60 km, and ~50-
60˚S and ~60 km. The zonal wind amplitudes of E3 occur at ~70-80˚S and

~60 km (~50-60˚S and ~60 km) with an amplitude of ~9 m/s on days 151-154, and ~9 m/s at ~70-80˚S and ~60 km (~50-60˚S and ~60 km) on days 201-204. The amplitude of E3 meridional wind hits ~13 m/s at ~60-70˚S and ~55 km (days 151-154) and ~16 m/s at ~60-70˚S and ~55 km (days 201-204), respectively.

EP flux of E3 is similar to that of E2. The instability and appropriate background wind at the mid-high latitudes between ~50 km and ~70 km dramatically amplify the propagation of E3, which is enhanced by the interaction near the critical layer (~29 h) and the positive refractive index region (Figures 6i and 6j). Notably, the strong instability and weak background wind at ~50-60˚S and ~60-70 km on days 151-154 provide sufficient energy for the propagation and amplification of EP flux into the lower atmosphere, and ultimately point toward the equator at 50 km. During days 201-204, the EP flux propagates into the lower atmosphere and gets amplified by interaction at the critical layer (~29 h). Besides, weak instability and weak background wind at ~50-60˚S and ~60-70 km provide the energy to amplify the E3 propagation. Figures 6c 6d indicate that the stronger the instability at ~50-60˚S and ~60-70 km, the stronger the temperature amplitude of E3. We believe that the background wind and instability at ~50-60˚S and ~60-70 km are the main reasons for the propagation and amplification of EP flux into the lower atmosphere.

For E4, the spectra appear at ~48.2 km and ~50-60°S on days 127-

130 and 213-216 when the eastward wavenumber -4 signal with the wave period of ~25 h and ~21 h (see Figures 7a, 7b). The corresponding temperature spatial structures of these E4 (i.e., ~25 h and ~21 h) are shown in Figures 7c, 7d. The temperature spatial structure of E4 shows an obvious amplitude bimodal structure at ~50-60˚S and ~50 km, and ~50-60˚S and ~60 km, with the maximum at ~50-60˚S and ~50 km. The maximum temperature amplitude of E4 occurs at ~50 km and ~50-60˚S with an amplitude of ~4 K on the days 127-130, and the other peak is ~3 K (~60-70˚S and ~60 km). The temperature amplitude of ~3 K occurs at ~50 km (~60 km) during days 213-216. The corresponding spatial structures of zonal wind and meridional wind of these E4 are presented in Figures 7e, 7f, 7g and 7h. The zonal wind spatial structure of E4 shows an obvious amplitude bimodal structure at ~50-60˚S and ~55 km, and ~60-70˚S and ~55 km, with the maximum at ~50-60˚S and ~55 km. The maximum zonal wind amplitude of E4 happens at ~50-60˚S and ~55 km with an amplitude of ~9 m/s on days 127-130, and the other peak is ~5K (~60-70˚S and ~55 km). The zonal wind amplitude of ~5 m/s occurs at ~50-60˚S (~60-70˚S) and ~55 km on days 213-216. The amplitude of E4 meridional wind reaches ~8 m/s at ~60-70˚S and ~55 km (days 127-130) and ~10 m/s at ~60-70˚S and ~55 km (days 213-216), respectively.

Diagnostic analysis for E4 on days 127-130 and 213-216 are shown in Figures 7i and 7j, respectively. The results demonstrate that E4 is

dramatically amplified by the mean flow instabilities at the middle-high latitudes between ~50 km and ~70 km. With EP flux propagating into the lower atmosphere, it finally propagates toward the equator at ~50 km. E4 is amplified and propagated by the wave-mean flow interaction near the critical layer (~25 h, ~21 h), and the positive refractive index region generates the promoting effect. The strong instability and weak background wind at ~50-60°S and ~60-70 km provide sufficient energy for the propagation and amplification of EP flux into the lower atmosphere during days 127-130. Besides, E4 obtains energy from weak instability and weak background wind at ~50-60°S and ~60-70 km on days 213-216, and it is amplified and propagated into the lower atmosphere. The background wind at ~50-60°S and ~60-70 km on days 127-130 is similar to on days 213-216, and the instability at ~50-60°S and ~60-70 km is stronger on days 127-30 than on days 213-216. According to Figures 7a and 7b, E4 absorbs sufficient energy to be amplified under the background conditions on days 127-130, and the temperature amplitude on 127-130 days is stronger.

**3.2 In the Northern Hemisphere**

Figure 8 shows that the observed maximum temperature amplitude appears at ~59.2 km and ~70-80°N for E1, and the E2 and E3 peaks at ~59.2 km and ~60-70°N. The maximum perturbation of E1 occurs on days 41-50 (with an amplitude of ~8K), while the remaining fluctuations occur on days 25-34 and 339-348. Besides, the strongest E2 occurs on days 69-

74 (with an amplitude of ~7 K), and the rest are distributed on days 25-30,
317-322 and 341-346. By contrast, the E3 maximizes on days 35-38 (with
an amplitude of ~3K), and also shows a peak on days 53-56. Based on the
study of the wave period in the SH for eastward wave, the periodic
variabilities of E1, E2 and E3 in the NH are also examined. The wave
period of E1 decreases from a maximum of ~118 h (days 25-34) to ~80 h
(days 41-50), indicating the instability of the wave period of E1 in the NH.
The E2 events occur on days 25-30, 69-74, 317-322 and 341-346, of which
the corresponding wave periods are ~36, ~53, ~52 and ~48 h, which are
stronger and more stable than E1. Besides, the wave period of E3 is
relatively stable at ~29 h and ~27 h. Thus, E2 and E3 wave periods are
more stable than E1.

The spectra, spatial (vertical and latitudinal) structures of temperature,
zonal and meridional wind, and diagnostic analysis of E1 are extracted
from the corresponding representative events (as shown in Figure 9).
Figures 9a and 9b show the observed spectra of E1 at ~59.2 km and ~70-
80°N on days 25-34 and 41-50, and the wave periods of locked
wavenumber -1 are ~118 h and ~80 h, respectively. The corresponding
temperature spatial structures of these E1 (~118 h and ~80 h) are shown in
Figures 9c, 9d. The temperature spatial structure of E1 shows an obvious
amplitude bimodal structure during days 25-34 at ~60-70°N and ~60 km,
and ~40-50°N and ~70 km, with the maximum at ~60-70°N and ~60 km.

On top of that, E1 also has bimodal structure on days 41-50 at ~60-70˚N and ~60 km, and ~60-70˚N and ~70 km. The strongest temperature amplitude of E1 occurs at ~60-70˚N and ~60 km with an amplitude of ~7 K on the days 25-34, and the other peak is ~4 K (~40-50˚N and ~70 km). The temperature amplitude of ~10 K occurs at ~60 km and ~60-70˚N during days 41-50, and the rest is ~8 K (~60-70˚N and ~70 km). The corresponding spatial structures of zonal wind and meridional wind of these E1 are illustrated in Figures 9e, 9f, 9g and 9h. The zonal wind spatial structure of E1 presents an obvious amplitude bimodal structure at ~70-80˚N and ~70 km, and ~50-60˚N and ~70 km. The zonal wind amplitude of ~13 m/s occurs at ~70-80˚N and ~70 km on days 25-34, and the rest is ~10 m/s (~50-60˚N and ~70 km). In addition, there is a weak peak of 9K during days 25-34 (~30-40˚N and ~70 km). The maximum zonal wind amplitude of E1 occurs at ~70-80˚N and ~70 km with an amplitude of ~19 m/s on days 41-50, and the other peak is ~13K (~50-60˚N and ~70 km). The amplitude of E1 meridional wind hits ~14 m/s at ~70-80˚N and ~70 km (days 25-34) and ~22 m/s at ~70-80˚N and ~70 km (days 41-50), respectively.

The diagnostic analysis results for E1 (in Figures 9i and 9j) suggest the dramatic amplification of E1 by the mean flow instabilities at the middle-high latitudes between ~50 km and ~70 km. With the propagation of EP flux into the polar lower atmosphere, it eventually propagates toward

the equator at ~50 km. The wave-mean flow interaction near the critical

layers (~118 h, ~80 h) amplifies and propagates E1, and the promoting

effect of the positive refractive index region amplifies E1. Furthermore, the

weak instability and strong background wind at ~40-50°N and ~60-70 km

generate the energy for the propagation and amplification of EP flux into

the polar lower atmosphere during days 25-34. The E1 obtains sufficient

energy from weak instability and suitable background wind on days 41-50

at ~40-50°N and ~60-70 km, and is amplified and propagated into the polar

lower atmosphere through the critical layer and positive refractive index

action. The background wind at ~40-50°N and ~60-70 km is stronger on

days 25-34 than on days 41-50, but their instability is similar, indicating

that stronger background winds might weaken E1 propagation and

amplification at the mid-northern latitudes. Our results show that E1

absorbs adequate energy to be amplified under the background conditions

during days 41-50, reflecting stronger temperature amplitude (see Figures

9a, 9b).

For E2, the spectra are at ~59.2 km and ~60-70°N on days 25-30 and

69-74 when the eastward wavenumber -2 signal with the period ~36 h and

~53 h (as shown in Figures 10a, 10b). The corresponding temperature

spatial structures of these E2 (i.e., ~36 h and ~53 h) are presented in Figures

10c, 10d. The temperature spatial structure of E2 demonstrates an obvious

amplitude bimodal structure at ~60-70°N and ~60 km, and ~60-70°N and

~70 km, with the maximum at ~60-70˚N and ~60 km. The maximum temperature amplitude of E2 occurs at ~60-70˚N and ~60 km with an amplitude of ~5 K on days 25-30, and the other peak is ~4 K (~60-70˚N and ~70 km). The temperature amplitude of ~9 K occurs on days 69-74 at ~60˚S and ~60 km, and the other peaks are ~7 K (~60-70˚N and ~70 km), ~5 K (~60-70˚N and ~50 km). The corresponding spatial structures of zonal wind and meridional wind of these E2 are shown in Figures 10e, 10f, 10g and 10h. The zonal wind spatial structure of E2 shows an obvious amplitude bimodal structure at ~60-70˚N and ~60 km, and ~40-50˚N and ~60 km, with the maximum at ~40-50˚N and ~60 km. The maximum zonal wind amplitude of E2 appears at ~60-70˚N and ~60 km (~40-50˚N and ~60 km) with an amplitude of ~6 m/s on days 25-30. Zonal wind amplitude occurs at ~40-50˚N and ~60 km with an amplitude of ~18 m/s on days 41-50, and the other peak is ~16 K (~60-70˚N and ~60km). The amplitude of E2 meridional wind reaches ~7 m/s at ~60-70˚N and ~70 km (days 25-30) and ~18 m/s at ~60-70˚N and ~60 km (days 41-50), respectively.

The diagnostic analysis of E2 on days 25-30 and 69-74 are shown in Figures 10i and 10j, respectively. Apparently, E2 is significantly amplified by the mean flow instabilities at the middle-high latitudes between ~40 km and ~70 km, with EP flux propagating into the polar lower atmosphere, and EP flux eventually propagate toward the equator at ~50 km. E2 is amplified and propagated by the wave-mean flow interaction near the critical layers

(~36 h, ~53 h), and the positive refractive index region provides the promoting effect. The weak instability and strong background wind at ~50-60°N and ~60-70 km provide the energy for the propagation and amplification of EP flux into the polar lower atmosphere during days 25-30. Moreover, E2 obtains sufficient energy from strong instability and suitable background wind at ~50-60°N and ~60-70 km on days 69-74, and it is amplified and propagated into the polar lower atmosphere. The background wind at ~50-60°N and ~60-70 km on days 127-130 is similar to on days 213-216, and the instability at ~50-60°S and ~60-70 km is stronger on days 127-30 than on days 213-216. Although the background wind at ~50-60°N and ~60-70 km is stronger on days 25-30 than on days 69-74, the instability at ~50-60°N and ~60-70 km is stronger on days 69-74 than on days 25-30. The temperature amplitude results indicate that E2 absorbs sufficient energy to be amplified under the background conditions on days 69-74, with stronger temperature amplitude on days 69-74 (Figures 10a, 10b).

Figures 11a and 11b show the observed spectra of E3 at ~59.2km and ~60-70°N on days 35-38 and 53-56, and the wave periods of locked wavenumber -3 are ~29 h and ~27 h, respectively. The corresponding temperature spatial structures of these E3 (i.e., ~29 h and ~27 h) are shown in Figures 11c, 11d. The temperature spatial structure of E3 shows an obvious amplitude bimodal structure at ~50-60°N and ~60 km, and ~50-

60˚N and ~70 km, with the maximum at ~50-60˚N and ~60 km. The strongest temperature amplitude of E3 occurs at ~60 km and ~50-60˚N with an amplitude of ~6 K on the days 35-38, and the other peak is ~5 K (~50-60˚N and ~70 km). The temperature amplitude of ~4 K occurs at ~60 km (~70 km) and ~50-60˚N during days 53-56. The corresponding spatial structures of zonal wind and meridional wind of these E3 are illustrated in Figures 6e, 6f, 6g and 6h. The zonal wind spatial structure of E3 shows an obvious amplitude bimodal structure at ~40-50˚N and ~70 km, and ~60-70˚N and ~70 km. The zonal wind amplitudes of E3 occur at ~40-50˚N and ~70 km with an amplitude of ~15 m/s on days 35-38, and ~12 m/s at ~60-70˚N and ~70 km. The maximum zonal wind amplitude of E3 appears at ~40-50˚N and ~70 km (~60-70˚N and ~70 km) with an amplitude of ~7 m/s (~6 m/s) on days 53-56. The amplitude of E3 meridional wind reaches ~22 m/s at ~50-60˚N and ~70 km (days 35-38) and ~12 m/s at ~60-70˚N and ~70 km (days 53-56), respectively.

Obviously, the instability and appropriate background wind at the mid-latitudes between ~50 km and ~70 km and the interaction near the critical layers (~29 h, ~27 h) dramatically amplify the propagation of E3 (see Figures 11i and 11j). The background wind is similar on days 35-38 and 53-56, and the former is relatively unstable. This finding indicates that the E3 in propagation is more likely to gather sufficient energy to be amplified on days 35-38. The instability and appropriate background wind

at the mid-high latitudes between ~50 km and ~70 km drastically amplify the propagation of E3, which is enhanced by the interaction near the critical layers (~29 h, ~27 h) and the positive refractive index region (Figures 11i, 11j). In particular, the strong instability and weak background wind at ~50-60˚N and ~60-70 km on days 35-38 generate sufficient energy for the propagation and amplification of EP flux into the lower atmosphere, and ultimately point toward the equator at 50 km. The EP flux propagates to the lower atmosphere during days 35-38, and it is amplified by interactions at the critical layer (~29 h). In addition, weak instability and weak background winds on days 53-56 at ~50-60˚N and ~60-70 km provide the energy to amplify E3 propagation. Combine with Figures 11c, 11d, the stronger the instability at ~50-60˚N and ~60-70 km, the stronger the temperature amplitude of E3. The results show that the instability on days 35-38 at ~50-60˚N and ~60-70 km are the primary reasons for the propagation and amplification of EP flux into the lower atmosphere.

**3.3 Comparison between SH and NH**

The observed latitude and maximum temperature amplitude for eastward planetary waves (i.e., E1, E2, E3, E4) decrease and weaken with increasing zonal wavenumber in the SH, reaching ~70-80˚S, ~60-70˚S, ~60-70˚S and ~50-60˚S, and ~10 K, ~9 K, ~6 K, and ~3 K, respectively. In addition, the occurrence date gets earlier with increasing zonal wavenumber. The temperature spatial structure demonstrates a bimodal-

peak structure (~50 and ~60 km), mainly located at ~50 km. The maximum zonal wind amplitudes of E1 and E2, E3 and E4 are almost the same, which are ~20 m/s and ~10 m/s, respectively. The maximum meridional wind amplitudes of E1, E2, E3 and E4 are ~17 m/s, ~27 m/s, ~16 m/s and ~11 m/s, respectively. The wave period of E1 tends to get shorter from 5 to 3 days; while E2 and E3 are close to ~40 h and ~30 h; and E4 remains at ~24 h. E1, E2, E3 and E4 are more favorable to propagation in the SH winter and are abruptly amplified by the mean flow instabilities at the middle latitudes between ~40 km and ~70 km. With the propagation of EP flux into the lower atmosphere, and it finally propagates toward the equator at ~50 km. In addition, the propagation of EP flux for E1 to the upper atmosphere might be influenced by the instability and background wind at the Antarctic ~50km.

The observed latitudes of E1, E2 (E3) decrease with increasing wavenumber in the NH, which are ~70-80°N, ~60-70°N and ~60-70°N. With bimodal-peak structure located at ~70 km, the temperature spatial structures of E1, E2 and E3 reach ~10 K, ~9 K and ~6 K, respectively. The maximum zonal wind amplitude for E1, E2 and E3 occur at ~50-80°N and ~70 km, and their amplitude are almost equal to ~18 m/s. The maximum meridional winds of E1, E2 and E3 occur at ~50-80°N and ~70 km with amplitudes of ~22 m/s, ~18 m/s and ~22 m/s, respectively. The wave period of E1 tends to be shorter from 5-3 days; and E2 and E3 are close to ~48 h

and ~30 h. In addition, E1, E2 and E3 are more favorable to propagation in the NH winter and are dramatically amplified by the mean flow instabilities at the middle latitudes between ~40 km and ~70 km, with the propagation of EP flux into the lower atmosphere and then toward the equator at ~50 km.

**4 Summary and Conclusions**

Based on the MERRA-2 temperature and wind observations in 2019, we present for the first time an extensive study of the global variation for eastward planetary wave activity, including zonal wave numbers of -1 (E1), -2 (E2), -3 (E3), -4 (E4) in the stratosphere and mesosphere. The temperature and wind amplitude and wave periods of each event were determined using 2-D least-squares fitting. Our study covered the spatial and temporal patterns of the eastward planetary waves in both hemispheres with a comprehensive diagnostic analysis on their propagation and amplification. The key findings of this study are summarized below:

1. The latitudes for the maximum (temperature, zonal and meridional wind) amplitudes of E1, E2, E3 and E4 decrease with increasing wavenumber in the SH and NH. The E1, E2, E3 and E4 events occur earlier with increasing zonal wavenumber in the SH. In addition, eastward wave modes exist during summer periods with westward background wind in both hemispheres.

2. The temperature spatial structures of E1, E2, E3 and E4 present a

double-peak structure, which is located at ~50 km and ~60 km in SH, ~60 km and ~70 km in SH. Furthermore, the lower peak is usually larger than the higher one.

3. The maximum (temperature, zonal and meridional wind) amplitude of E1, E2 and E3 decline with rising zonal wavenumber in the SH and NH. The maximum temperature amplitude in the SH are slightly larger and lie lower than those in the NH. In addition, the meridional wind amplitude are slightly larger than the zonal wind in the SH and NH.

4. The wave period of the E1 mode ranges 3-5 days in both hemispheres, while the period of E2 mode is slightly longer in the NH (~48 h) than in the SH (~40 h). The periods of E3 in both SH and NH are ~30 h, while the period of E4 is ~24 h.

5. The eastward planetary wave is more favorable to propagate in the winter hemisphere and is drastically amplified by the mean flow instabilities and appropriate background winds at polar region and the middle latitudes between ~40 km and ~80 km. Furthermore, the amplification of planetary waves through wave-mean flow interaction occurs easily close to its critical layer. In addition, the direction of EP flux ultimately points towards the equator.

6. The strong instability and appropriate background wind in the lower layer of the Antarctic region might generate adequate energy to promote the E1 propagation and amplification to the upper atmosphere.

Overall, this study demonstrated how the background zonal wind in

the polar middle atmosphere affects the dynamics of eastward planetary

waves in the polar middle atmosphere.

*Data availability.* MERRA-2 data are available at http://disc.gsfc.nasa.gov.

*Code availability.* Code is available at http://hdl.pid21.cn/21.86116.7/04.99.01720.

*Author contributions.* LT carried out the data processing and analysis and wrote the manuscript. SYG and XKD contributed to reviewing the article.

*Competing interests.* The authors declare that they have no conflict of interest.

*Acknowledgements.* This work was performed in the framework of the Space Physics Research (SPR). The authors thank NASA for free online access to the MERRA-2 temperature reanalysis.

*Financial support.* This research work was supported by the National Natural Science Foundation of China (41704153, 41874181, and 41831071).

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

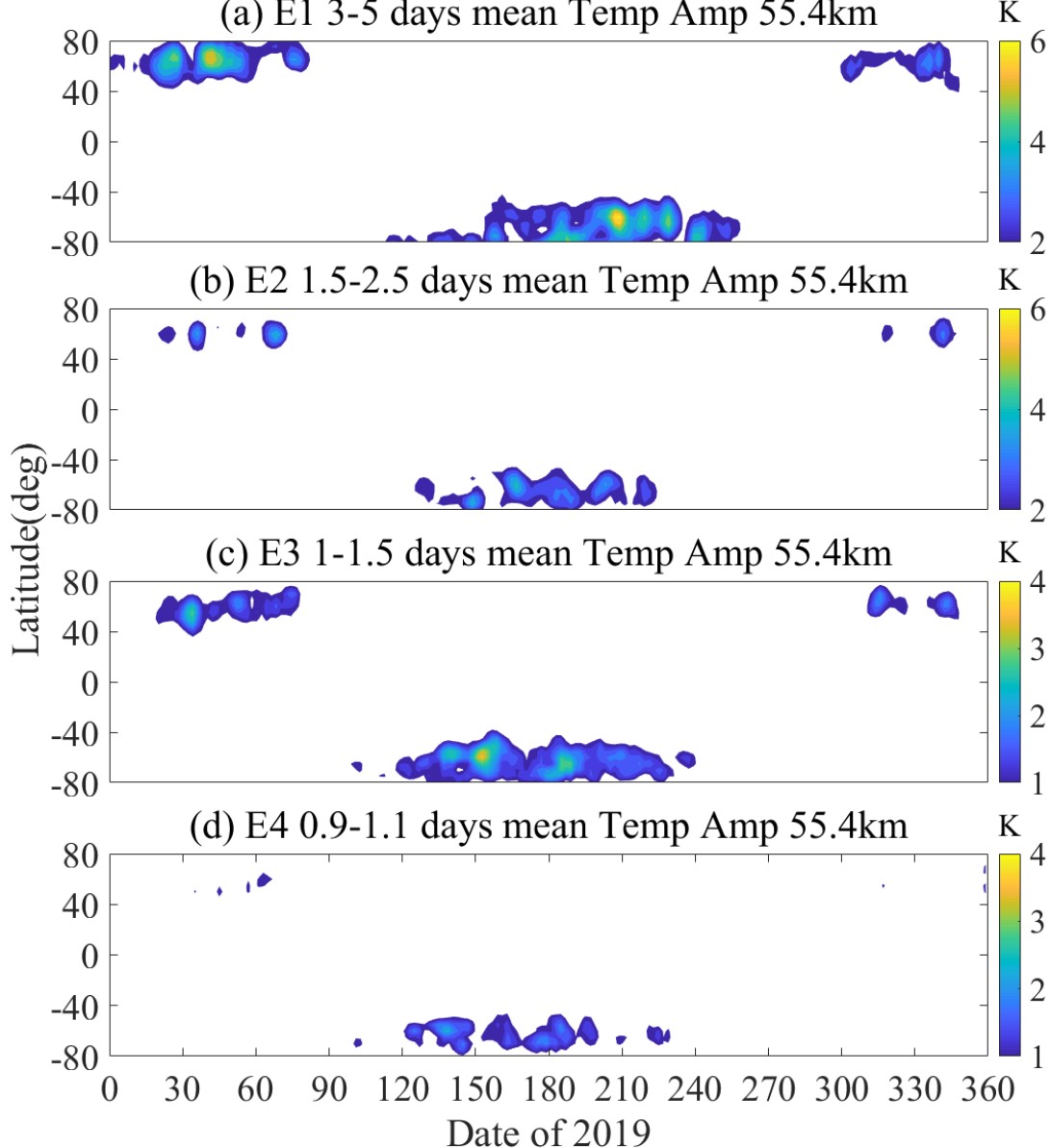

**Figure 1.** The global latitude-temporal variation structures of the (a) E1, (b) E2, (c) E3 and (d) E4 planetary waves during 2019. White areas represent small amplitude data (corresponds to the right color bar).

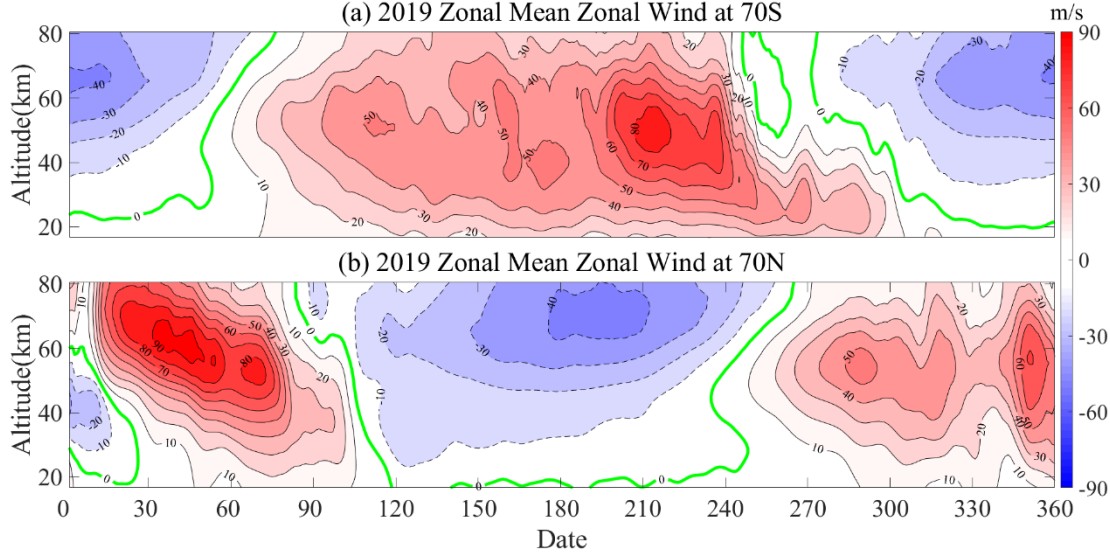


**Figure 2.** The zonal mean zonal wind variations of (a) 70°S and (b) 70°N during 2019.
The dotted line represents eastward wind, the solid line represents westward wind, and
the green solid line is 0 m/s.

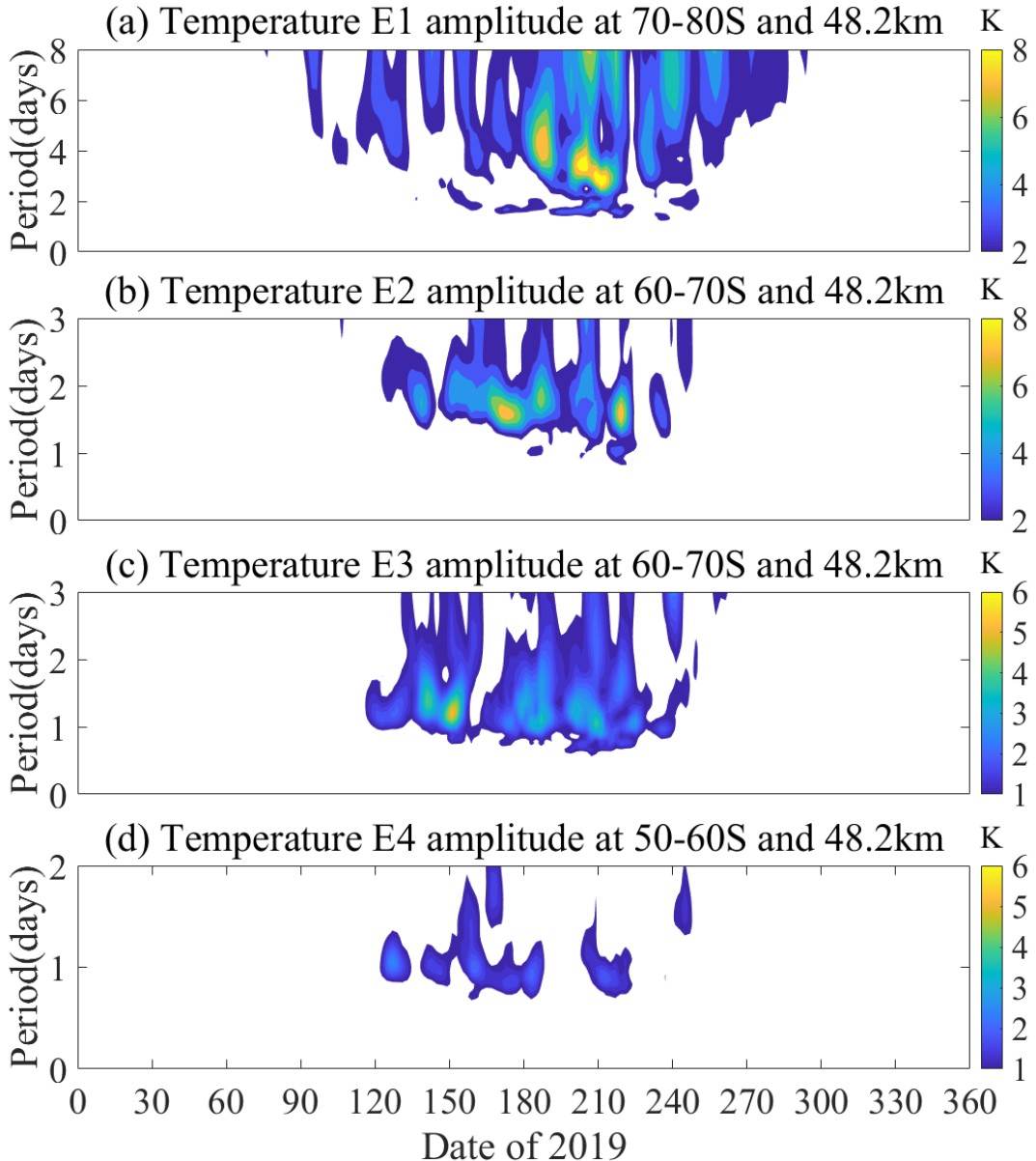


**Figure 3.** The temporal variations of (a) E1, (b) E2, (c) E3 and (d) E4 QTDWs during

2019 austral winter period.

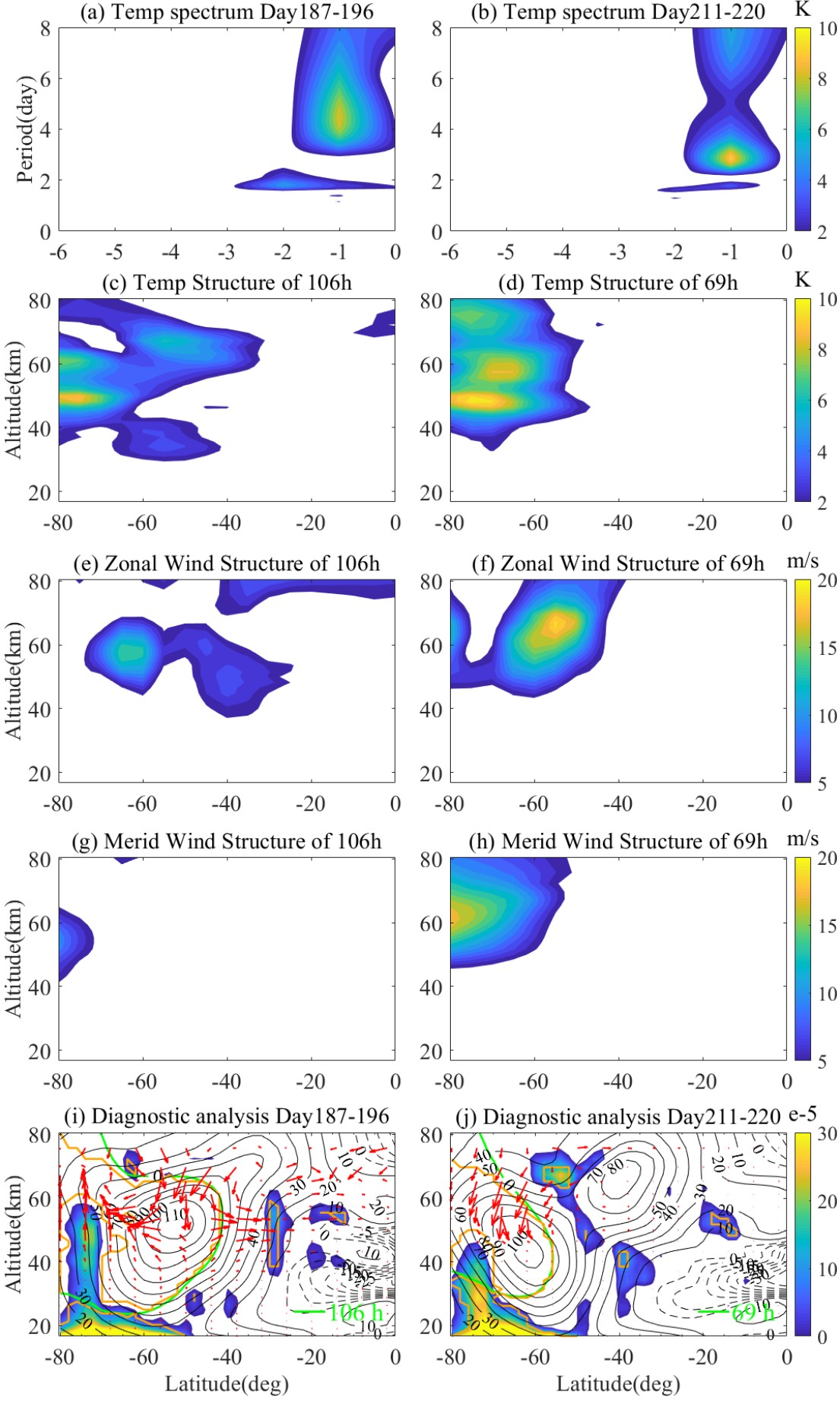


**Figure 4.** The (a, b) spectra, (c, d) temperature spatial structures, (e, f) zonal wind
spatial structures, (g, h) meridional wind spatial structures, and (i, j) diagnostic analysis
of the E1 typical events during 2019 austral winter period. The MERRA-2 temperature
data observations at 48.2 km and 70-80°S during days 187–196 (Figure 4a), 211–220
(Figure 4d) are utilized, respectively. In the diagnostic analysis of E1 events, the blue
shaded region is instability, the red arrow is EP flux, and the green line is critical layer.
The green line represents critical layers of E1 with the natural period. Regions enclosed
by orange solid lines are characterized by the positive refractive index for the E1.

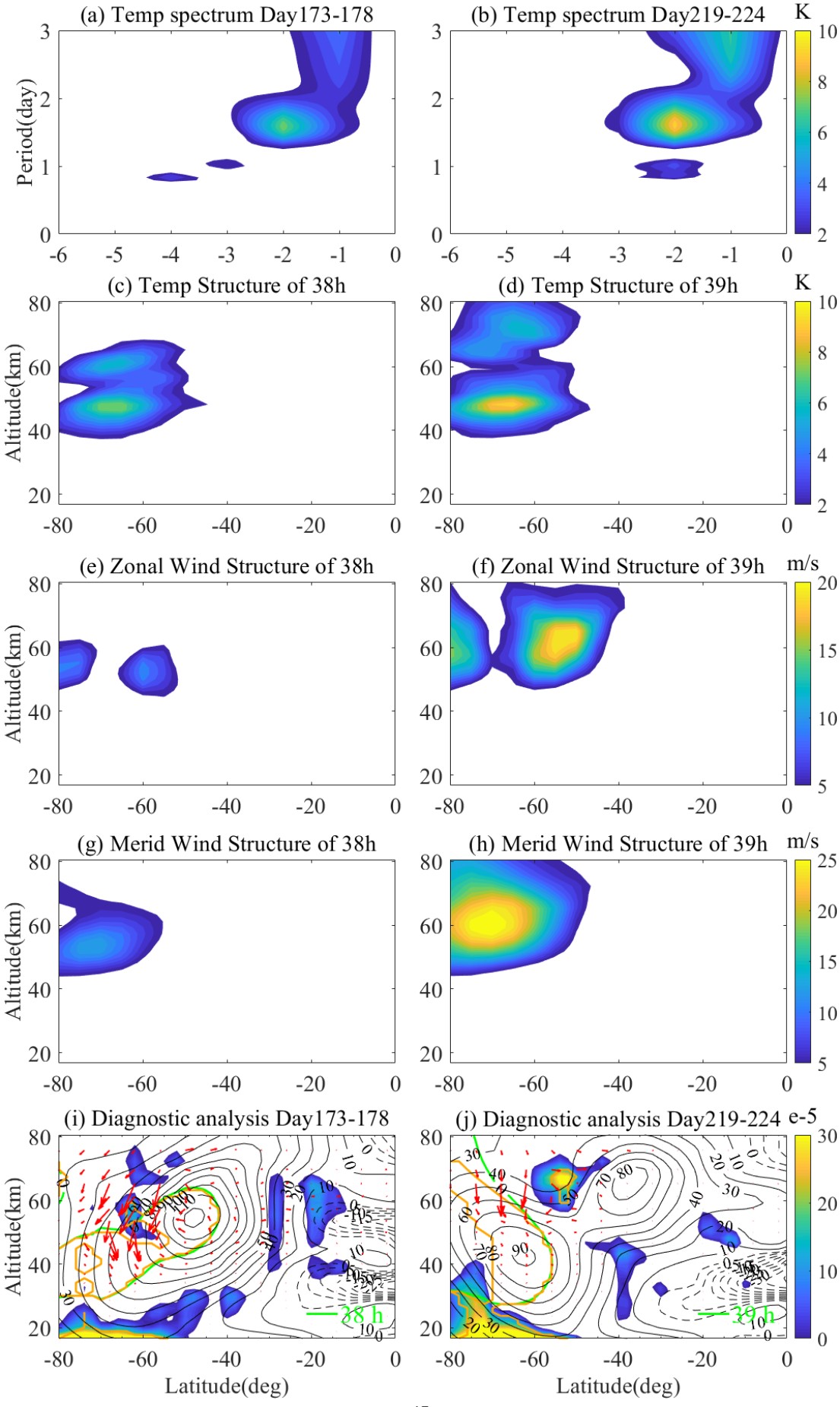


**Figure 5.** Same as Figure 4 but for E2 during the 2019 austral winter period.

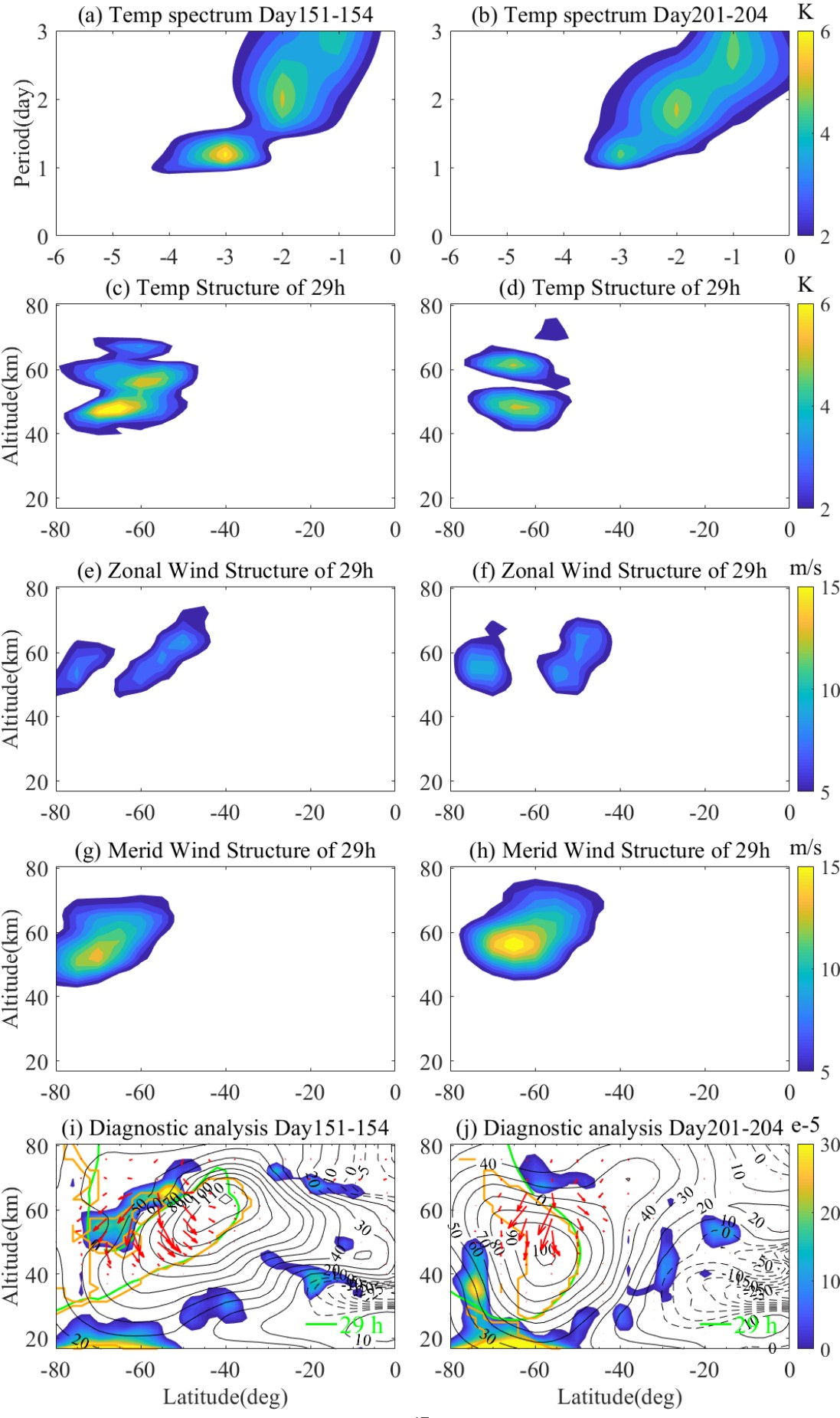

**Figure 6.** Same as Figure 4 but for E3 during the 2019 austral winter period.

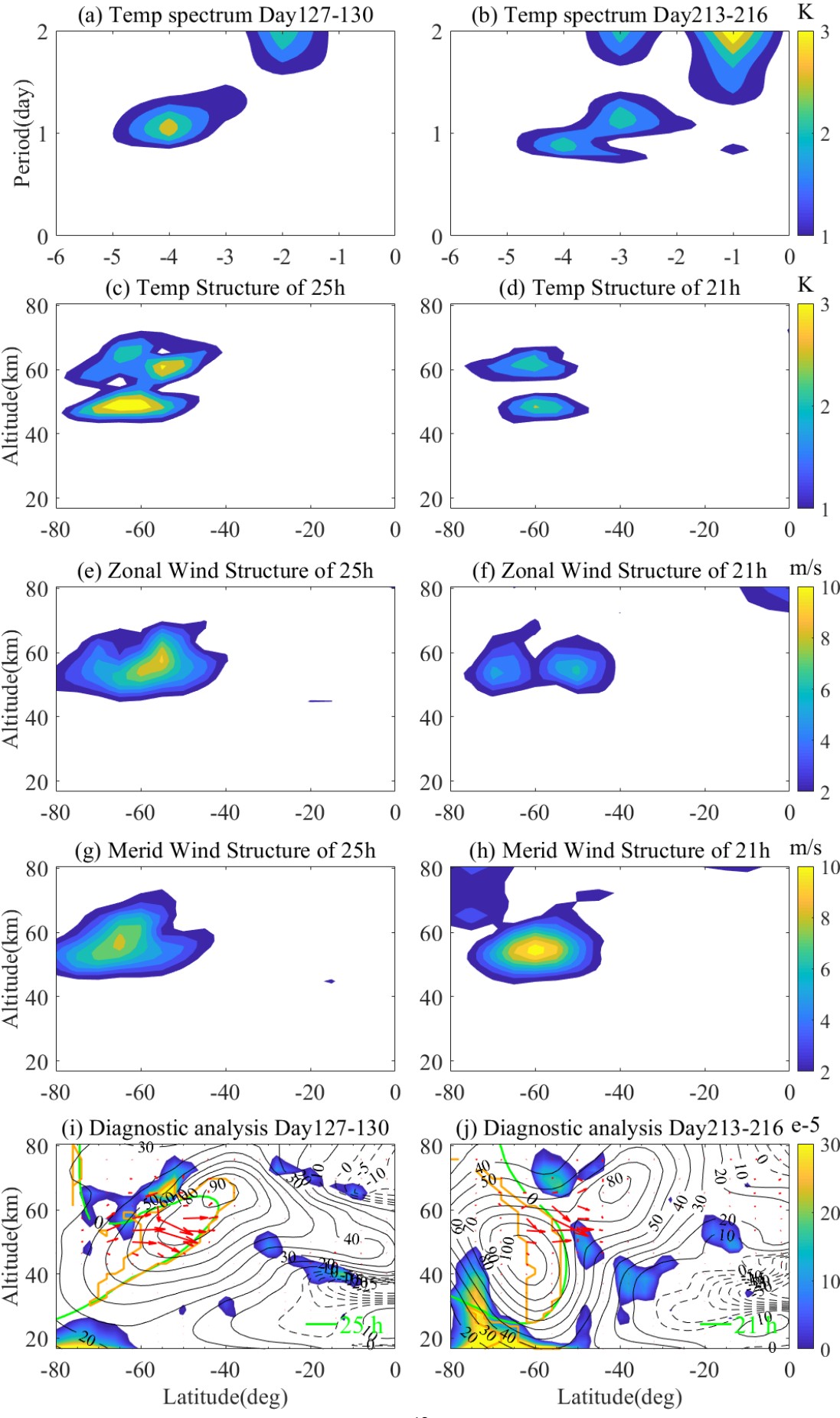

**Figure 7.** Same as Figure 4 but for E4 during the 2019 austral winter period.

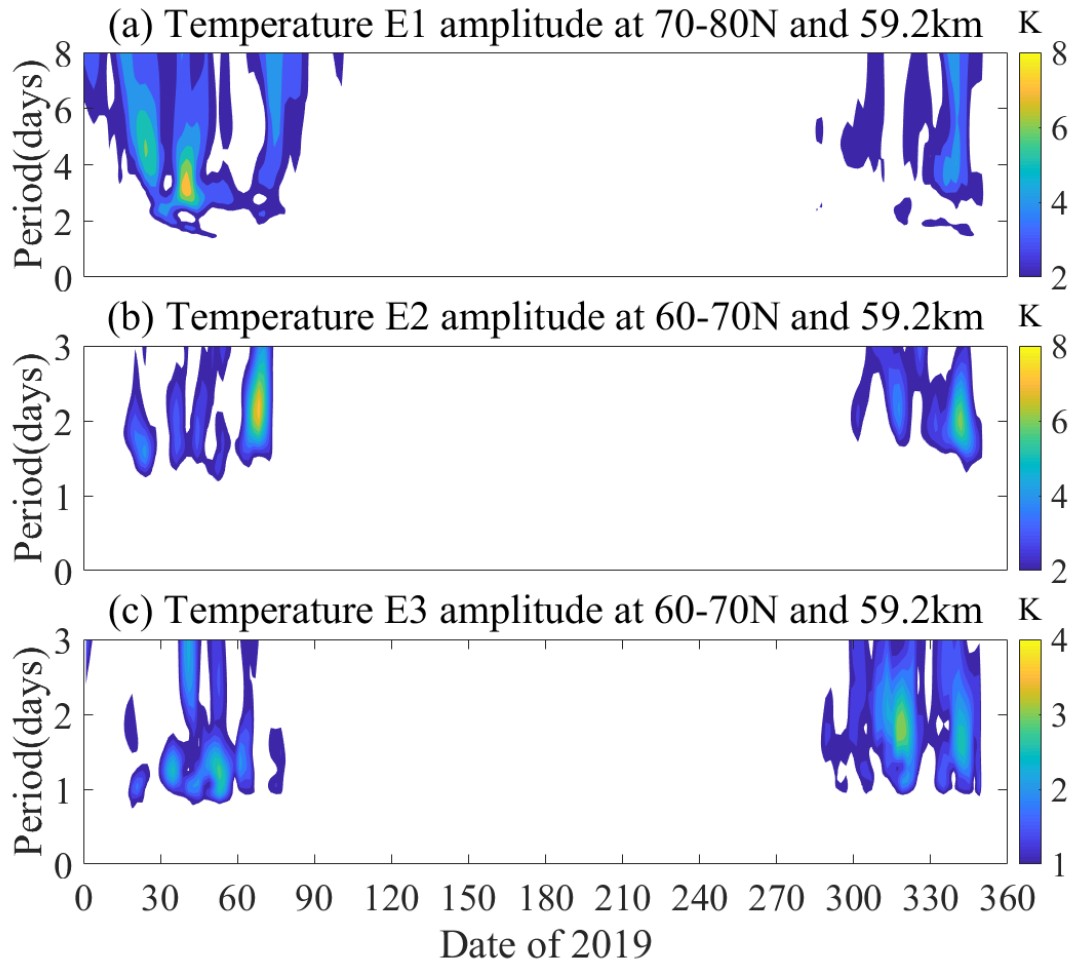


**Figure 8.** The temporal variations of (a) E1, (b) E2 and (c) E3 QTDWs during the 2019
boreal winter period.

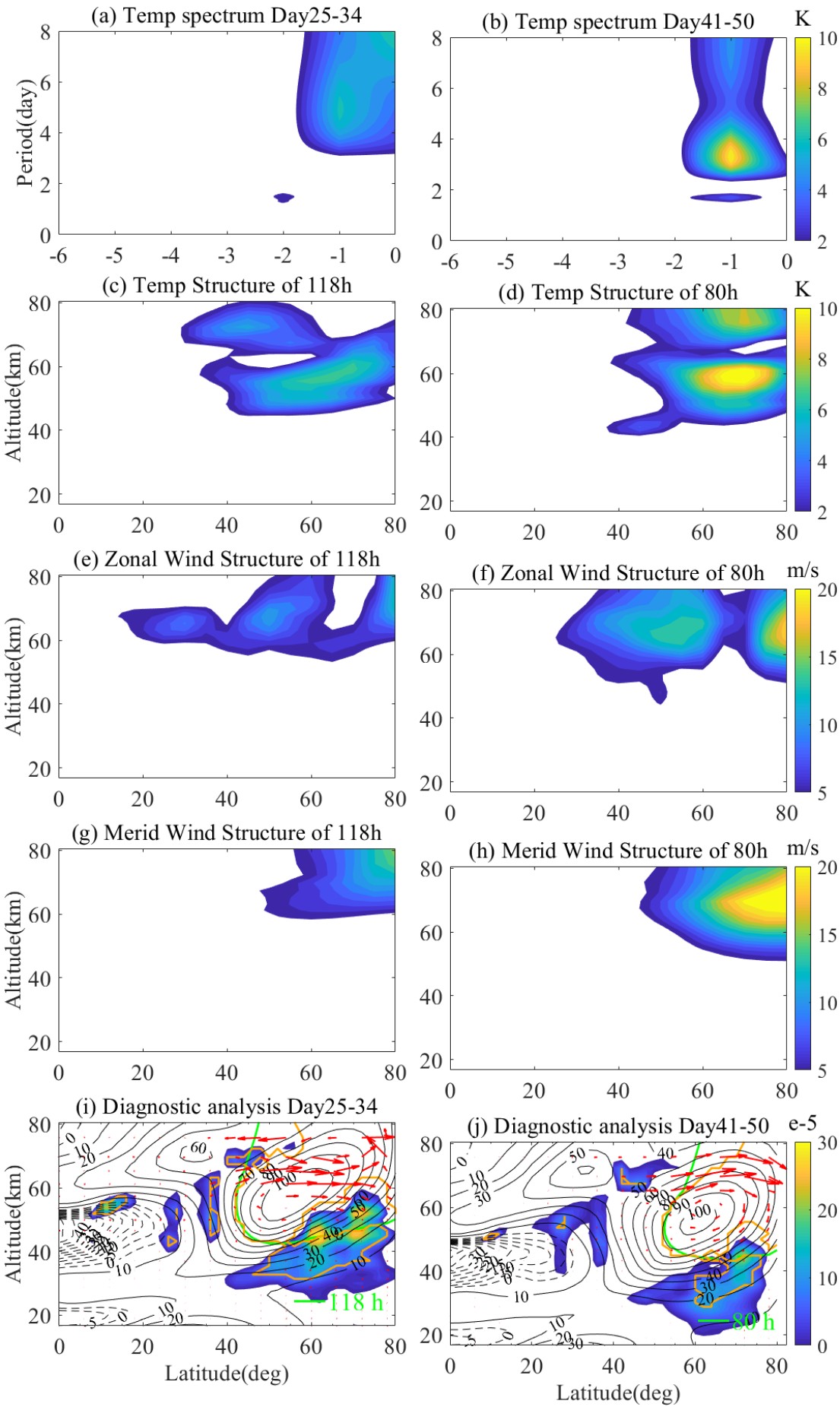

**Figure 9.** The (a, b) spectra, (c, d) temperature spatial structures, (e, f) zonal wind
spatial structures, (g, h) meridional wind spatial structures and (i, j) diagnostic analysis
of the E1 typical events during 2019 boreal winter period. The E1 events at 48.2 km
and 70-80°N were obtained from the MERRA-2 reanalysis.

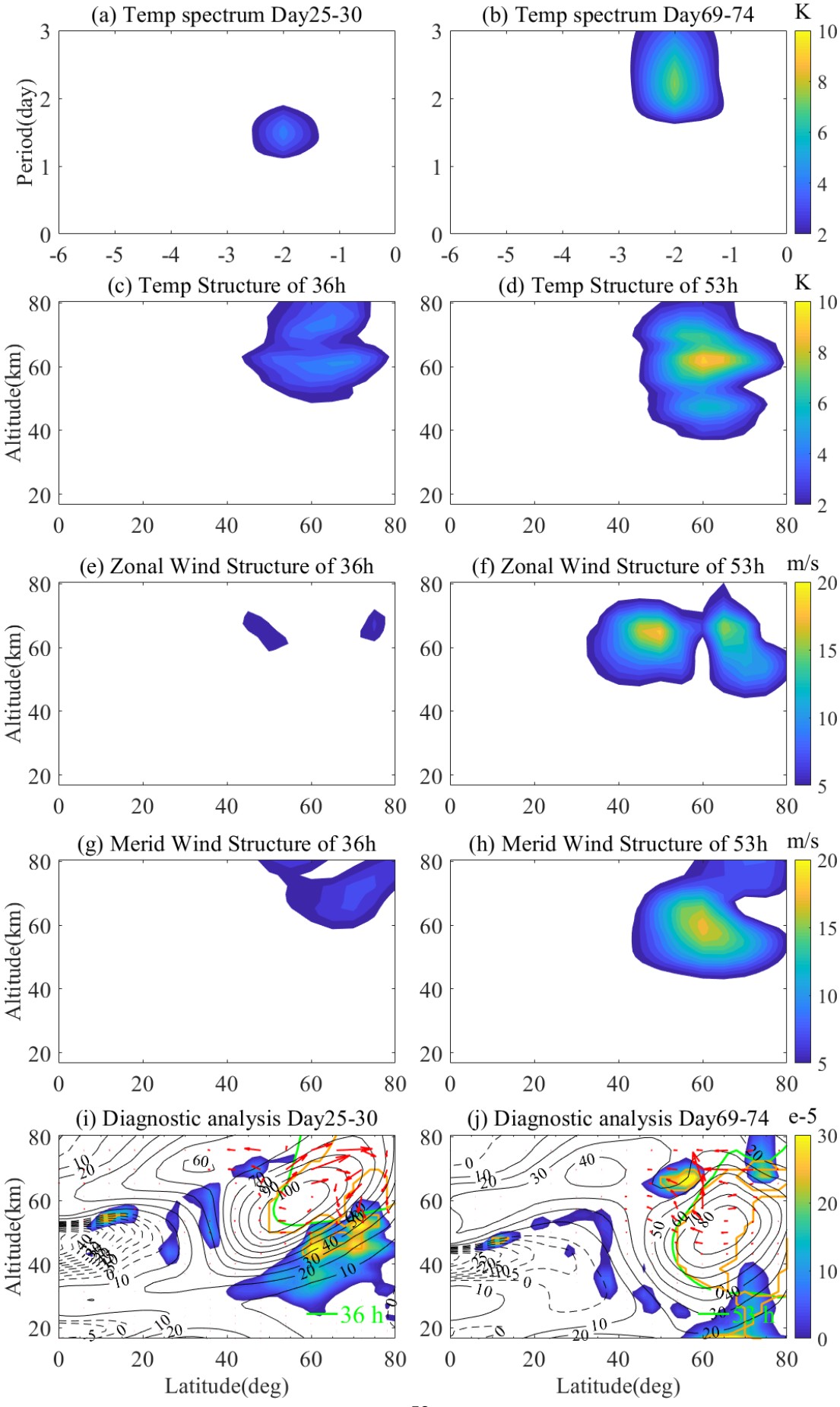


**Figure 10.** Same as Figure 9 but for E2 during the 2019 boreal winter period.

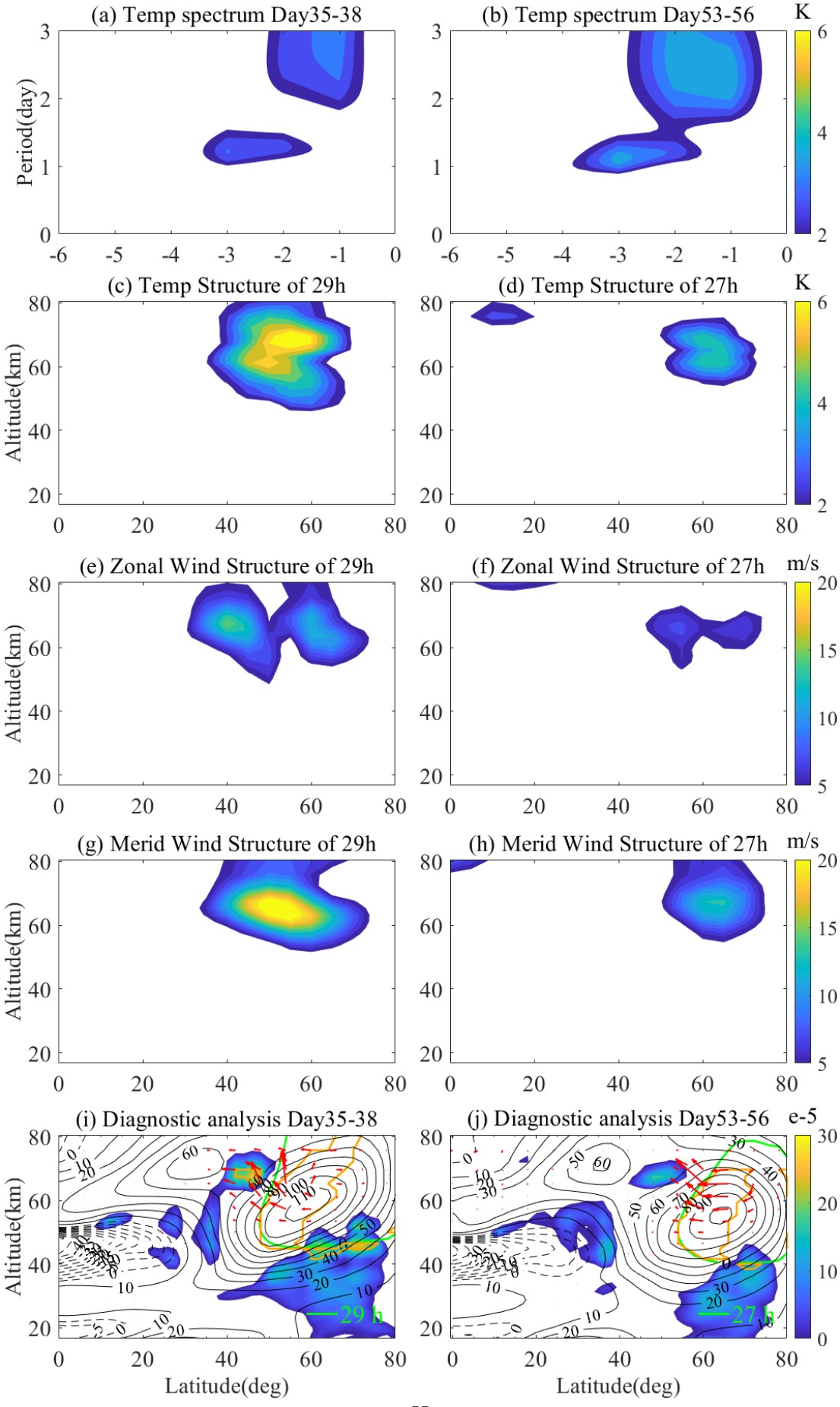

**Figure 11.** Same as Figure 9 but for E3 during the 2019 boreal winter period.