# Peer review of "Eastward-propagating planetary wave in the polar"

_Atmospheric Chemistry and Physics, 2021_

## Author Comment (AC2)

We thank the reviewers and editors for their constructive comments on our manuscript. The manuscript is revised thoroughly by considering all the comments. Besides, Figures 2, 4, 5, 6, 7, 9, 10, and 11 have been updated to make the results clearer. Besides, the language is polished by the Edit Springs English editing service. Our responses to every comment are listed below with blue.

**Response to Anonymous Referee 2**

Specific comments

Due to the extensity and focus of the study, I would appreciate an adoption of Open Science approaches to allow reproduce the extensive analysis in this study (e.g. Laken, 2016). In particular, I would recommend any kind of willingness of the authors to publish the code allowing to reproduce the figures in the paper. There are multiple ways how to proceed, either to allow the access upon request or via portals allowing to assign Digital Object Identifier (DOI) to the research outputs, e.g. ZENODO. I think it could enhance the quality and reliability of this publication. In the end, this publication might be motivating for future middle atmosphere studies.

All the MATLAB codes and data used for the analysis in this study are available at http://hdl.pid21.cn/21.86116.7/04.99.01293.

Authors should consider using a diverging colormap in Figure 2 to clearly differentiate between positive and negative values (Zeller and Rogers, 2020).

In the revision, the eastward and westward winds are distinguished by dotted and solid lines, respectively. Besides, the zero line is also highlighted, which makes it much easier to differentiate eastward and westward wind.

To improve Figure 4 and its successors deserve improvements in terms of description and graphical representations of EP fluxes. The size of the arrows may need to be increased. Using vector figures instead of raster ones may help to differentiate details as well.

More descriptions on the analysis results are added in the revision. The arrow size for EP flux have also been increased.

Is there any reason why only one year was analysed? Would you expect any differences between reanalysis datasets in terms of your results? The same one-year analysis may be done based on the ERA5 reanalysis.

This paper focuses the seasonal variations of the eastward wave modes in the polar stratosphere. In fact, we also checked the wave events during other years, and we found that the four waves modes, e.g., E1, E2, E3 and E4, are all representative during 2019. Thus, the analysis results during 2019 are presented. The inter-annual variations and the corresponding mechanisms would be investigated next.

It is a pity that we do not have the whole ERA5 dataset currently. And it may take a much longer time to download the whole ERA5 dataset to reproduce all the figures in the manuscript. We downloaded the temperature data during July and August of 2019 and found that the analysis results are roughly consistent with MERRA2. In the future, we would perform more comparisons between these two reanalysis datasets.

[Figure]

Figure S1. The comparison of the temperature spectra with eastward wavenumber 1 between (a) MERRA2 and (b) ERA5.

Technical comments

l58 switch position of "long-term" and "observed"
Revised in the revision.

References

Laken, B. A. (2016). Can Open Science save us from a solar-driven monsoon? Journal of Space Weather and Space Climate, 6, A11. http://doi.org/10.1051/swsc/2016005
Zeller, S., and D. Rogers (2020), Visualizing science: How color determines what we see, Eos, 101, https://doi.org/10.1029/2020EO144330. Published on 21 May 2020.

---

## Author Response (AR1)

We thank the reviewers and editors for their constructive comments on our manuscript. The manuscript is revised thoroughly by considering all the comments. Besides, Figures 2, 4, 5, 6, 7, 9, 10, and 11 have been updated to make the results clearer. The reference format has also been updated. Our responses to every comment are listed below with blue.

**Response to Anonymous Referee 1**

1) There are too many grammar mistakes and some of the sentences are difficult to understand. I have list some of them in the detailed list. I did not list all the errors. The language need to be polished more.

The manuscript is polished by the EditSprings English editing service before the resubmission, and more detailed descriptions are added in the revision.

2) The figures are not explained well. Especially Figures 4-7 and 9-11, the i and j figures need to be better explained. I have a hard time following the discussion of these figures.

More descriptions on the analysis results are added in the revision.

Specific comments:

Line 94-95, Further research by Palo et al. (2007). Also this sentence is confusing. Do you mean eastward Q2DW was produced by the nonlinear interaction?

Yes, that is correct. I have revised the following sentence in the manuscript to reflect this: " Beyond the knowledge about nonlinear interactions between migrating tides and Q2DWs (Palo et al., 1999), further investigation has confirmed that E2 Q2DW is coupled by nonlinear planetary wave and tides in the mesosphere and lower thermosphere (Palo et al., 2007)." Meanwhile, we need to explain that E2 generation may be caused by the nonlinear interactions of MLT region, or it may be caused by the instability of polar region. E2 occurs at different altitudes and latitudes in these two modes.

Line 113, This sentence is confusing. How can 'propagation height' be limited to high latitudes? Do you mean eastward waves are limited?

The text has been corrected as follows: "Lu et al. (2013) found that eastward planetary wave propagation is limited to the winter high latitudes probably because the negative refractive indices equatorward of ~45°S result in evanescent wave characteristics."

Figure 1, please specify what the white areas represent, missing data, data too small, etc., either in the caption or the text. Without this information, I can not follow the discussion involved these figures.

White areas in Figure 1 represent small signals (corresponds to the right color bar). This is made clear in the revison.

Line 215-216, First, please describe what are showed in Figure 3. Second, please specify why these latitude bands are chosen for these waves. I assume that these are where the peaks are found in Figure 2, but please state it clearly in the paper.

Figure 3 shows the span of period for every planetary wave mode. And the latitudes and altitudes chosen in Figure 3 are where the corresponding wave peak. These two

points are made clear in the revision.

Line 228-229, Please justify the statement 'the PWs E1-E4 have similar phase speeds' by, for instance, explaining how the phase speeds are calculated, and specifying what are values of the phase speeds.

The calculation of phase speed has been added in the revision. And the values of the phase speed are also clearly stated in the revision.

$$c = -\upsilon_0 \cos\left(\frac{\varphi\pi}{180}\right)\Big/ sT$$

where $c$ is the phase speed; $\upsilon_0$ is the equatorial linear velocity; $\varphi$ is the latitude; $s$ is the zonal wavenumber and $T$ is the wave period.

Line 230-232, Please explain more detailly what are shown in each figure of Figure 4. Specifically, what are 'temperature structure'? From the context I can guess it's the amplitude of the wave in temperature, but it should be clearly stated in the paper, instead of leaving it for the readers to figure out.

Temperature structure means the vertical and latitudinal distribution of the wave amplitude in temperature for the eastward waves. Every figure is stated more clearly in the revision.

Figures 4-8, and 9-11, Please explain what the filled colors in figures i and j are. Without this information, I can not follow the discussion associated with these figures.

Figure i and j are exhibited to show the propagation and amplification of every wave (below). The blue shaded region represent instability, and the red arrow represent EP flux. The green line represents critical layers. Regions enclosed by orange solid lines are characterized by the positive refractive index. More descriptions on the analysis results are added in the revision.

Technical comments.

Line 25-26: seasonal variations of the critical layers generated by the background wind
Revised in the revision.
Line 45: 'The seasonal variations' to 'Seasonal variations'
Revised in the revision.
Line 54: 'a maximum amplitude' to 'amplitudes'
Revised in the revision.
Line 58: 'the W3' to 'W3'
Revised in the revision.
Line 59: delete 'those of'
Revised in the revision.
Line 62: citation format needs correction, 'the wave' to 'wave'
Revised in the revision.
Line 67: 'confused' to 'indistinguishable','during the SSWs period' to 'during SSWs'
Revised in the revision.

Line 68-69: to 'Then periods of W3, W4 and W2 vary between …., respectively

Revised in the revision.

Line 71, Than the tropics?

No. W2 can be observed in global satellite datasets, showing weaker amplitude than W3 and W4 in the NH and SH. This is stated more clearly in the revison.

Line 72, 'dominated' to 'modulated'

Revised in the revision.

Line 76, 'the SSWs period' to 'the SSWs'. Revise all the terms in the manuscript

Revised in the revision.

Line 81, It should be 'planetary waves' or 'planetary wave activities'

Revised in the revision.

Line 82, change to ' with periods of nearly 2 and 4 days'

Revised in the revision.

Line 86-88, 'In addition, planetary waves of …..have been found to have the same phase speeds as…'

Revised in the revision.

Line 109, 'proposed' to 'found'

Revised in the revision.

Line 120, 'eastward propagation wave' to 'eastward propagating wave'

Revised in the revision.

Line 127, 'In Section 2, …',

Revised in the revision.

Line 132, 'Section 3.3 compares and analyzes ..'

Revised in the revision.

Line 136, Any reasons for choosing these windows (i.e.,10, 6, 4, 4 days)?

We chose analysis windows that are ~2-3 times as long as the wave periods without other specific reasons.

Line 144, 'amplitude of wave (not wavenumber)'

Revised in the revision.

Line 170-176, These definitions of terms should be moved after Eq.2. Also when defining the terms of an equation with 'where', the first letter should not be capitalized, and it should not be a new paragraph (for example Line 182, 187)

Revised in the revision.

We thank the reviewers and editors for their constructive comments on our manuscript. The manuscript is revised thoroughly by considering all the comments. Besides, Figures 2, 4, 5, 6, 7, 9, 10, and 11 have been updated to make the results clearer. The reference format has also been updated. Our responses to every comment are listed below with blue.

**Response to Anonymous Referee 2**

Specific comments

Due to the extensity and focus of the study, I would appreciate an adoption of Open Science approaches to allow reproduce the extensive analysis in this study (e.g. Laken, 2016). In particular, I would recommend any kind of willingness of the authors to publish the code allowing to reproduce the figures in the paper. There are multiple ways how to proceed, either to allow the access upon request or via portals allowing to assign Digital Object Identifier (DOI) to the research outputs, e.g. ZENODO. I think it could enhance the quality and reliability of this publication. In the end, this publication might be motivating for future middle atmosphere studies.

Code and data availability All the MATLAB codes and data used for analysis of this study are available at http://hdl.pid21.cn/21.86116.7/04.99.01293.

Authors should consider using a diverging colormap in Figure 2 to clearly differentiate between positive and negative values (Zeller and Rogers, 2020).

The eastward and westward winds are distinguished by dotted and solid lines, respectively.

To improve Figure 4 and its successors deserve improvements in terms of description and graphical representations of EP fluxes. The size of the arrows may need to be increased. Using vector figures instead of raster ones may help to differentiate details as well.

More descriptions on the analysis results are added in the revision. The red arrows for EP flux have also been redrawn.

Is there any reason why only one year was analysed? Would you expect any differences between reanalysis datasets in terms of your results? The same one-year analysis may be done based on the ERA5 reanalysis.

We present the E1, E2, E3 and E4 events in 2019 because the amplitudes and periods of these four events are typical. Different reanalysis datasets may cause differences for amplitude, because different reanalysis datasets (ERA5, etc.) are based on different data and algorithms, but the final results for our research should be similar. Thus the results do not change much even if ERA5 reanalysis data were applied to our study.

[Figure]

Technical comments

l58 switch position of "long-term" and "observed"
Revised in the revision.

References
Laken, B. A. (2016). Can Open Science save us from a solar-driven monsoon? Journal of Space Weather and Space Climate, 6, A11. http://doi.org/10.1051/swsc/2016005
Zeller, S., and D. Rogers (2020), Visualizing science: How color determines what we see, Eos, 101, https://doi.org/10.1029/2020EO144330. Published on 21 May 2020.

---

## Author Response (AR2)

We thank the reviewers and editors for their constructive comments on our manuscript. The manuscript is revised thoroughly by considering all the comments. Besides, Figure 2 have been updated to make the results clearer. Our responses to every comment are listed below with blue.

**Response to Anonymous Referee 2**

The manuscript was significantly improved. However, I still list further comments below.

Specific comments

I checked the code availability. I could download the code but could not uncompress the RAR file. Providing the code in a freeware format like ZIP or others is highly recommended.

All the MATLAB codes and data used for analysis of this study are available at http://hdl.pid21.cn/21.86116.7/04.99.01720 (ZIP format).

The readability of Figure 2 significantly improved. However, the authors should still reconsider using a diverging colormap in Figure 2 to clearly and intuitively differentiate between positive and negative values (see guideline e.g. in Figure 6 from Crameri et al. 2020). The "viridis" colormap as used in Fig. 2 represents a sequential colormap where the lightness value should increase monotonically with values without centric value though. I let the editor decide on this issue.

Figure 2 is replotted before the resubmission, which is shown below by Figure S1.

[Figure]

Figure S1. The zonal mean zonal wind variations of (a) 70°S and (b) 70°N during 2019. The dotted line represents eastward wind, the solid line represents westward wind, and the green solid line is 0 m/s.

l84 A sentence from the revised manuscript "Some recent studies have discovered significant eastward planetary..." does not correspond with the sentence from the manuscript with revised changes: "Recent studies have found significant eastward planetary waves..." Hopefully, this is the only case. The authors should comment on

this discrepancy.

The manuscript was polished by an English Service before the resubmission. Some of the changes were not included in the final version during the last submission. We double checked the revised manuscript this time, and the tracked version is not consistent with the clean version this time. Sorry for our mistakes.

I still miss a justification why E1, E2, E3 and E4 events only in 2019 are presented in the manuscript. It can be connected with future outlook in the end.

We analyzed the MERRA-2 dataset at the beginning, and found that all the four wave modes are clearly exhibited. We thus presented the analysis results during 2019. The MERRA-2 datasets during other years (2013-2020) are analyzed later. We found that the analysis results during 2019 are representative for all the four wave modes. The analysis during other years exhibits similar seasonal features as that presented during 2019. And the results in the southern hemisphere are shown below as an example. In the future, we would like to study the inter-annual variations of the eastward planetary waves in detail, which may be presented in a new paper.

[Figure]

Figure S2. The temporal variations of E1 during 2013-2020 in the SH.

[Figure]

Figure S3. The temporal variations of E2 during 2013-2020 in the SH.

[Figure]

Figure S4. The temporal variations of E3 during 2013-2020 in the SH.

[Figure]

Figure S5. The temporal variations of E4 during 2013-2020 in the SH.

Technical comments
l75 missing word (verb?) behind "Gu et al. (2019) have..."
Revised in the revision.

References
Crameri, F., Shephard, G.E. & Heron, P.J. The misuse of colour in science communication. Nat Commun 11, 5444 (2020). https://doi.org/10.1038/s41467-020-19160-7

---

## Author Response (AR3)

Thanks for the valuable suggestion.

**Response to editor**

The only remaining point that concerns me is the following. You have stated in your response to Referee 2 that you have considered the years 2013-2020 but have included results only from 2019 because -- on the basis of your consideration of 2013-2020 -- the results from 2019 are representative of all years. But, as far as I can tell, you have not included any statement on that in the paper itself.

My recommendation is that you include such a statement in the paper itself -- it will remove uncertainty and add to the value of your paper. (If you had not checked that the results presented for 2019 were representative of many other years then this would have been a major weakness.)

You could add a statement like:

"We have presented results only for the year 2019 but have considered the entire period 2013-20 and find that the results presented are representative of all years in this range.'

If you have some very strong reason for NOT doing that then please give that in your response.

We do have analyzed all the eastward waves during 2013-2020, which are shown below. It is clearly shown that the eastward planetary waves during 2019 are representative. Only the analysis results during 2019 is presented mainly due to the similar seasonal features. We also note that the wave events exhibit interannual variabilities, which would be another interesting topic and will be investigated in the future.

The statement suggested by the editor is added in the revision.

[Figure]

Figure S1. The temporal variations of E1 during 2013-2020 in the SH.

[Figure]

Figure S2. The temporal variations of E2 during 2013-2020 in the SH.

[Figure]

Figure S3. The temporal variations of E3 during 2013-2020 in the SH.

[Figure]

Figure S4. The temporal variations of E4 during 2013-2020 in the SH.